# Molecular Basis of the Anticancer and Antibacterial Properties of CecropinXJ Peptide: An In Silico Study

**DOI:** 10.3390/ijms22020691

**Published:** 2021-01-12

**Authors:** Francisco Ramos-Martín, Nicola D’Amelio

**Affiliations:** Unité de Génie Enzymatique et Cellulaire UMR 7025 CNRS, Université de Picardie Jules Verne, 80039 Amiens, France

**Keywords:** antimicrobial peptide, antibiotic resistance, anticancer, esophageal carcinoma, molecular dynamics, biophysics, sequence alignment

## Abstract

Esophageal cancer is an aggressive lethal malignancy causing thousands of deaths every year. While current treatments have poor outcomes, cecropinXJ (CXJ) is one of the very few peptides with demonstrated in vivo activity. The great interest in CXJ stems from its low toxicity and additional activity against most ESKAPE bacteria and fungi. Here, we present the first study of its mechanism of action based on molecular dynamics (MD) simulations and sequence-property alignment. Although unstructured in solution, predictions highlight the presence of two helices separated by a flexible hinge containing P24 and stabilized by the interaction of W2 with target biomembranes: an amphipathic helix-I and a poorly structured helix-II. Both MD and sequence-property alignment point to the important role of helix I in both the activity and the interaction with biomembranes. MD reveals that CXJ interacts mainly with phosphatidylserine (PS) but also with phosphatidylethanolamine (PE) headgroups, both found in the outer leaflet of cancer cells, while salt bridges with phosphate moieties are prevalent in bacterial biomimetic membranes composed of PE, phosphatidylglycerol (PG) and cardiolipin (CL). The antibacterial activity of CXJ might also explain its interaction with mitochondria, whose phospholipid composition recalls that of bacteria and its capability to induce apoptosis in cancer cells.

## 1. Introduction

The aim of this work is to characterize the mechanism of action of cecropinXJ (CXJ), an anticancer peptide (ACPs) also acting as antimicrobial peptide (AMP) towards bacteria [1,2,3]. As anticancer, CXJ has been shown to target mitochondria of apoptotic cells [4,5,6], meaning that it can act as a cell penetrating peptide (CPP) [7,8] but also as mitochondrial penetrating peptide (MPP) [9,10,11]. Notably, CXJ is one of the very few AMPs reported to be active against esophageal cancer (EC) [4]. In the following, we briefly introduce the reader to the problems associated with EC, the importance of targeting mitochondria and the mechanism of action of peptides targeting biological membranes. In our study, we will show how CXJ can selectively interact with biomimetic models of cancer cells and how the similarity of mitochondrial membrane composition to that of bacteria could explain the additional antibacterial properties of CXJ.

EC is a highly aggressive lethal malignancy causing thousands of deaths annually [12,13]. It is classified into two main histopathological subtypes: esophageal squamous cell carcinoma (ESCC) and esophageal adenocarcinoma (EAC). Although they share some similarities, they differ in cellular origin, incidence, epidemiology, and molecular signatures. ESCC is the predominant subtype in the world, but EAC is more prevalent in developed countries [14]. Additionally, the incidence of ESCC tends to decrease in favor of EAC, which is among the fastest growing malignancies [12,13]. Gastroesophageal reflux is the most documented risk factor for EAC, which can gradually evolve from the premalignant Barrett’s esophagus [15]. EAC is usually detected at an advanced stage with only a 15% 5-year survival rate. This cancer is molecularly heterogeneous and poorly understood. It displays the highest amount of mutation frequency [16], making patient-tailored therapies ineffective. Only two agents have been approved for treatment in the metastatic setting [17]. Unfortunately, current oncological protocols, involving neoadjuvant chemotherapy and surgical management, generally fail to improve patient outcomes. Despite surgery [18], there is a high incidence of recurrence even in combination with chemo and radiotherapy [19,20]. In summary, the major obstacles for developing new therapeutic approaches is the lack of understanding of the molecular genetic drivers of EAC and its high inter-tumor heterogeneity. The lack of good physiological model systems is another issue, although recent advances (e.g., organoids [21]) might open new scenarios.

Evidence in favor of a direct implication of mitochondria in oncogenesis has been accumulating [22,23,24] and highlights the importance of specific mitochondrial-targeting chemotherapies [25]. Cancer cells tend to disable the mitochondrial pathway of apoptosis by suppressing signals that can cause mitochondrial outer membrane permeabilization. In particular, EC development [26,27] and Barrett’s esophagus disease [28] are linked to mitochondrial dysfunction. Targeting mitochondria to induce apoptosis of carcinogenic cells is, therefore, an appealing strategy applicable to multiple forms of resistant cancers [29]. The induction of apoptosis is currently exploited by chemotherapy and radiotherapy, which cause DNA damage leading to its activation via intrinsic and/or extrinsic pathways, both involving mitochondrial membrane permeabilization. This results in the release of species (such as cytochrome c) activating the apoptosome [30].

MPPs, a special class of CPPs able to target mitochondria, have displayed remarkable properties for medical applications [31]. They have been shown to cause apoptosis in cancer cells [6] and to enter the cell without necessarily damaging the plasmatic membrane [7,8,32]. Once inside, they can target the mitochondrion acting as membrane-disrupting agents or inducing apoptosis with different mechanisms [31]. MPPs are particularly interesting as anticancer drugs, because, as most ACPs, they promise to overcome the main limitations of chemotherapy: drug resistance and non-specificity. Their specificity is due to an intrinsic difference between the plasma membranes of most human cells and that of cancer cells, in terms of charge and fluidity. As opposed to the essentially uncharged human cell membranes, cancer cells tend to be more negatively charged because of a large amount of sialic-acid-rich glycoproteins, phosphatidylserine (PS) or heparan sulfate [33]. Inside the cell, MPPs display additional selectivity for mitochondria, whose membranes are peculiar in terms of lipid composition (containing cardiolipin, CL) and potential [34]. The capability of MPPs to enter the mitochondrion accounts for their reduced sensitivity to resistance. First, mitochondrial DNA is not able to produce drug degrading enzymes; second, the mitochondrial membrane protects internalized peptides from efflux pumps [35]. Third, the absence of a complex mitochondrial DNA repairing system amplifies the toxicity of drugs [36], which can be delivered by conjugation to MPPs [36,37,38].

CXJ is a cationic AMP isolated from larvae of *Bombyx mori* belonging to the cecropin-B family. Most importantly, it is one of the few peptides with activity against EC [4] and also targets other cancer types such as gastric [4,5,39] or hepatocellular carcinoma [6]. Despite its potent anticancer activity, its mechanism of action is unknown, although it does not seem to affect the plasma membrane, suggesting that its action relies on other mechanisms [5]. CXJ is known to target mitochondria and induce apoptosis via the mitochondrial-mediated caspase pathway. It also downregulates anti-apoptotic Bcl-2, increases ROS species, affects the expression of cytoskeleton proteins [5], and disrupts mitochondrial membrane potential, leading to the release of cytochrome c. Similar to many other cecropins [40], CXJ displays a potent antibacterial action against a wide spectrum of gram-positive and gram-negative bacteria such as *Enterococcus faecalis*, *Staphylococcus epidermidis*, *Klebsiella pneumoniae*, *Acinetobacter baumannii*, *Shigella flexneri*, *Shigella sonnei*, *Staphylococcus aureus*, among others [2]. What is more, CXJ has a low hemolytic and cytotoxic activity, and it has been shown that amidation of its C-terminus improves its antibacterial properties and further reduces its low hemolytic action [3]. Finally, it has been shown to act as an antifungal agent against several organisms such as *Penicillium digitatum*, *Magnaporthe grisea* and, to a minor extent *Botrytis cinerea*, *Penicillium italicum* [1,2].

The mechanism by which cecropins recognize and interact with membranes is dictated by their primary sequence and by the composition of the target membrane. Differences in phospholipid composition of the plasma membrane of bacteria, cancer cell or mitochondria determine whether the peptide can affect, penetrate or disrupt its target. Antibacterial peptides generally recognize the bacterial membrane, rich in PG, PE and CL. The co-existence of apoptotic and antibacterial properties in CXJ is not surprising if one considers the similarity shared by mitochondrial and bacterial membranes [41,42]. ACPs mostly recognize PS, while the mechanism of penetration of CPPs is based on the presence of arginine residues coupled to the flip-flop motion of phospholipid in the bilayer [43,44]. All these data demonstrate that a description at atomic level of the interaction of cecropins with the membranes of cancer cells, bacteria, and mitochondria is key for developing optimized peptides for targeted therapies.

In summary, the importance of unraveling the unknown mechanism of action of CXJ is apparent when considering that: (i) CXJ is among the very few peptides active against esophageal cancer; (ii) it displays poor cytotoxicity and hemolytic activity; (iii) it has demonstrated potent activity against several ESKAPE bacteria; (iv) it interacts with biological membranes and can act as CPP and MPP. In this work, we use property-sequence alignment [45] to highlight conserved motifs involved in the biological activity of CXJ and all-atom molecular dynamics (MD) to unveil the very first steps of the interaction of CXJ with a variety of biomimetic membranes, representing eukaryotic and cancer cells, bacteria and fungi.

## 2. Results

### 2.1. Property-Sequence Alignment of CXJ

Figure 1A,B shows the sequence-related (SR) family created using the CXJ sequence as template. SR families can be generated by ADAPTABLE webserver [45], using sequence alignment among peptides with a defined biological activity. In our case, we selected peptides with both anticancer and antimicrobial activities and a minimal inhibitory concentration (MIC) lower than 10 μM. The resulting SR family is composed of 17 peptides, also displaying as a whole a variety of extra activities including antiviral (11.1% against HIV, HSV or Junin virus), antifungal (38.9%) or antiparasitic (22.2%). These percentages refer to available experimental data and have to be considered as minimum percentages (for example all peptides might be antiviral, but the antiviral activity has not been studied for all) [45].

A closer inspection to the SR family of Figure 1B reveals the presence of well conserved motifs, namely: RWK, KKIEK and GIVKAGPA. In these motifs, amino acids can be replaced by closely related ones in terms of charge of polarity. In particular, K can be substituted by R, E by D and V by I or L (Figure 1A).

The secondary structure of each peptide of the family can be predicted by PSSpred [46], which provides a DSSP (Define Secondary Structure of Proteins) [47] even for peptides lacking a known PDB structure (Figure 1C). Clearly two helices can be distinguished in most cases, separated by the “GP” part of the GIVKAGPA motif. From here on, we will refer to these helices as helix I (residues 1–22) and helix II (residues 25–37). Notably, proline residues are able to interrupt helical structures, while the small side chains of the flanking alanine and glycine residues are likely to allow multiple different relative orientations between the two helices.

### 2.2. CXJ Can Form an Amphipathic Helix but Remains Unstructured in Solution

In the hypothesis of a helical conformation for CXJ, the analysis of the helical wheel clearly shows that, while the N-terminus (residues 1–22) can form an amphipathic helix (Figure 2A), the C-terminus (22–37) has a larger proportion of hydrophobic residues. The predicted structure by I-TASSER [48] (Figure 2B) displays two long helices separated by a bend (A22-G23-P24) at the level of P24. This structure is in agreement with the predicted structure based on DSSP analysis (Figure 1C). A similar structure is also found in the Satpdb database (entry 12223 [49]).

MD simulation shows the predicted conformation is quickly lost in solution (Figure 2C), reproducing experimental data of the literature, describing an unstructured peptide which acquires 41.0% of helical content only in the presence of anionic sodium dodecyl sulfate (SDS) micelles (58.5% in the case of the amidated form) [3].

### 2.3. CXJ Does Not Interact with Phosphatidylcholine Membranes

In phosphatidylcholine (PC) bilayers, headgroups do not contain H-bonding donors, and electrostatic effects predominate. These include electrostatic attraction between positively charged choline groups and negatively charged phosphate moieties and electrostatic repulsion arising from both choline-choline and phosphate-phosphate contacts. However, the steric hindrance of the N-methyl groups markedly reduces in magnitude these interactions. As a consequence, despite the zwitterionic character of PC headgroups, electrostatic repulsion between phosphate moieties makes the membrane relatively fluid as demonstrated by the low values of its melting temperature T_m_ [51,52].

Although CXJ has been shown to interact with biomimetic membranes [3], we did not observe any kind of interaction with 1-palmitoyl-2-oleoyl-glycero-3-phosphocholine (POPC) bilayers (Figure 3A). This might indicate that the reported interaction with negatively charged SDS micelles [3] is mostly driven by electrostatics, keeping into account that CXJ is expected to display a global charge of +7 at physiological pH. The affinity for negatively charged membranes also explains its antibacterial activity [53]. The absence of interaction with POPC membranes is coherent with the exhibited low hemolytic properties (2% at 200 µM) [2] and low cytotoxicity [53]. PC headgroup can in fact be considered as a model for eukaryotic membranes, mostly composed of phosphatidylcholine in the outer leaflet [54,55]. In order to confirm this result, we tested the interaction of CXJ with POPC membranes containing 30% of cholesterol (CHO), which better represent the membrane of mammal cells. The most representative snapshot of such simulation is shown in Figure 3B. Once again, the structure is lost (see contact map in Appendix A), and no significant interaction with the membrane is observed (see Appendix A).

### 2.4. CXJ Specifically Recognises PS and PE Headgroups, Exposed in Apoptotic Cancer Cells

#### 2.4.1. The Effect of CXJ on PS-Containing Membranes

As opposed to the essentially uncharged mammalian cell membranes, cancer cell membranes tend to be more negatively charged because of a large amount of sialic-acid-rich glycoproteins, PS or heparan sulfate [33]. Metastatic cells can reduce membrane cholesterol content to increase the membrane fluidity and plasticity [56,57,58], a process that may enhance the exposure of lipids commonly found in the inner leaflet [59]. Cancer cells can also increase their cholesterol content depending on the changes in metabolism induced by carcinogenic and angiogenic processes [60,61,62]. Apoptotic cells tend to expose PS, a phospholipid normally found in the inner leaflet of the membrane. This phenomenon, called externalization, intervenes in a wide variety of biological processes, including activation of B-cells and platelets [63]. Its capital importance is demonstrated by its evolutive conservation as an “eat-me” signal and used by apoptotic cells to induce phagocytosis by professional and nonprofessional phagocytes in a noninflammatory manner (efferocytosis) [63]. A specific recognition of PS is probably the reason why some cecropins can act as anticancer agents, while displaying low hemolytic activity and toxicity to healthy cells [1,3,64,65]. Their eukaryotic origin explains their selectivity.

In our MD simulation, we modeled the external leaflet of an apoptotic cell by a mixture of PC, PS and CHO. A representative snapshot of such interaction is shown in Figure 3C. Contrarily to what observed with POPC (Figure 4A), the contact map calculated along the last 250ns of the simulation (Figure 4B) shows that CXJ retains a U-shaped structure approaching the N and C termini. The peptide interacts with the membrane mainly by means of the amphipathic helix I, while helix II displays a much higher degree of freedom. This is consistent with the small percentage (41.0%) of helical content experimentally observed by circular dichroism in the presence of anionic membrane models [3]. During the simulation, CXJ does not penetrate completely in the bilayer but interacts strongly and frequently.

In order to get insight into the nature of such interactions, we calculated the distribution function of each membrane N/O atom types from each N/O atom of CXJ along the simulations. Subsequently, we extracted the maximum of the function in the distance range compatible with H-bonding or salt bridges. The graph that we obtain is a measure of the occurrence of each polar interaction (Figure 5A and Appendix A for all PS containing membranes). The occurrence of polar contacts immediately reveals that CXJ has a net preference for PS (yellow) over PC (black) and CHO (red). This could be due to the negative charge of this lipid, attracting the positively charged CXJ, whose global charge is +7 at physiological pH.

A closer analysis of the data reveals that CXJ recognizes the carboxylate atoms of PS by means of the terminal NH_2_ groups of arginine residues in positions 1, 13 and 16. An important role for arginine residues has also been reported for other CPPs [8,66,67,68]. Such binding appears to occur much more frequently than other interactions driven by electrostatics, like those between the NH_3_^+^ groups of lysine side chains and the membrane phosphates (when calculating the total occurrence, the sum of the contributions from each of the two equivalent oxygen atoms O13A and O13B of the serine carboxylate should be considered). This is probably due to the fact that binding can take place in a bidentate fashion. Such an interaction might explain how CXJ could act as a cell CPP and exploit lipid flip-flops to be transported on the opposite side of the target membrane, while remaining anchored to the lipids [69,70]. The interactions of lysine side chains with phosphate groups also contribute to the binding and can be established with either POPC or 1-palmitoyl-2-oleoyl-sn-glycero-3-phospho-L-serine (POPS).

Interestingly the nitrogen atom of the tryptophan in position 2 makes a frequent H-bond with the hydroxyl of cholesterol. When the distribution function of lipid acyl chain is calculated taking apolar moieties of CXJ as a reference for the evaluation of van der Waals interactions (Figure 6A and Appendix A for all PS containing membranes), it becomes apparent that the N-terminus can insert its hydrophobic side chains and in particular that of the tryptophan W2. The presence of tryptophan is known to contribute to the uptake efficiency of CPPs, while its position in the sequence modulates it [71,72].

When cholesterol is removed from POPC/POPS/CHO (35/35/30%) bilayers (i.e., POPC/POPS (50/50%) membranes, Figure 3D), its absence results in a reduction in polar (Appendix A) and van der Waals (Appendix A) contacts with the bilayer. Finally, membranes containing only POPS (Figure 3E) maintain the described polar contacts (Appendix A), but the peptide is much less able to penetrate (Appendix A), probably due to the rich network of interactions among lipids and the strong electrostatic attraction at the level of the negatively charged membrane surface.

#### 2.4.2. The Effect of CXJ on PE Membranes

PE is a phospholipid present in the plasma membrane of both eukaryotes and prokaryotes, but in mammalian cells, it is generally found in the inner leaflet. In apoptotic cells, the membrane asymmetry is lost resulting in its exposure with PS on the outer leaflet of many different cancer cells [73,74,75,76], including EC [75]. PE gets also exposed on the surface of irradiated cells [77].

PE protonated amine group is able to form H-bonds as donor with the phosphate and carbonyl oxygen atoms of adjacent PE molecules. These H-bonds replace those between PE and water and strengthen inter-lipid contacts [78,79,80]. Under physiologically relevant conditions, PE amino groups are fully protonated, and their positive charges are capable of both attracting negatively charged groups and forming H-bond interactions. The relatively high transition temperatures of PE bilayers are the result of multiple contributions arising from electrostatic attraction between positively charged amino groups and negatively charged phosphate moieties, electrostatic repulsion arising from both amino-amino and phosphate-phosphate contacts, as well as H-bonding interactions and van der Waals contacts in the hydrophobic regions of the lipid bilayer [78,81].

Overall, CXJ remains quite structured on the surface of 1-palmitoyl-2-oleoyl-sn-glycero-3-phosphoethanolamine (POPE) all along the MD simulation and maintains head to tail contacts as in the case of POPS (see the contact map in Figure 4C). Figure 3F shows that CXJ interacts with PE membranes mostly by means of the amphipathic helix I. This is a non-obvious finding as PE membranes are globally neutral and expose positively charged amine groups that could repel the positively charged CXJ.

The analysis of polar contacts (Figure 5B) shows that E9, D17 and to a lesser extent the terminal carboxylate interact frequently with the amine of PE headgroup, while R1,13 and K3,6,10 make salt bridges with the more interior phosphate moieties of the membrane. Such an interaction has been previously reported for CPPs [44] as a means to transpass the bilayer. The analysis of apolar contacts (Figure 6B) reveals that in this case the side chain of W2 can be deeply inserted in the membrane core, confirming the previously reported importance of this residue [71,72].

### 2.5. CXJ Interacts with Bacterial Biomimetic Membranes and Each of Their Pure Components

The outer leaflet of the cytoplasmic membrane in gram-positive bacteria often contains anionic phospholipids such as PG and CL (as it is the case of *Staphylococcus aureus* [82] or *Staphylococcus epidermidis* [83]). In most gram-negative bacteria, such as *Pseudomonas aeruginosa* [84], *Acinetobacter baumannii* [85], *Escherichia coli* [86] or *Klebsiella pneumoniae* [87], PE is the major phospholipid present at all stages of the growth. PG is also present as the second most abundant component [88,89] and CL might be present too. It should be noted that bacteria can change phospholipid ratios as a response to environmental conditions [88]. Relevant factors are the growing phase, the availability of nutrients in the growing media, or the cultivation temperature [89,90,91].

This environment-dependent variability [82,83,84,85,86,87,88,89,90,91] and the intrinsic variety in the phospholipid composition found across the bacterial kingdom make it difficult to simulate all the different types of phospholipid ratios. We, therefore, opted for simulating systems containing pure components (PE, PG or CL, Figure 3G–J), able to highlight key interactions due to a single phospholipid. However, the preference of CXJ for a specific phospholipid and the presence of inter-lipid interactions can only be studied in mixtures. In particular, we studied various combinations of PE, PG and/or CL, in order to reproduce different types of bacterial membranes. Among these mixtures, we chose compositions similar to those found in *E. coli* [86] or *K. pneumoniae* [87], involving a higher amount of PE. Our data (e.g., mixtures in Figure 5) show that a clear preference can be established for specific phospholipids, which might explain resistant mechanisms based on changes in membrane lipid composition.

As we have already discussed the interaction with PE, in the following, we will analyze the interaction with PG, CL, PE/PG (70/30%) and PE/PG/CL (67/27/6%).

#### 2.5.1. The Effect of CXJ on PG Membranes

The hydroxyl groups of PG have a potential to form intermolecular H-bonds as was observed in monolayers, bilayers and different model membranes [92,93,94]. These inter-lipid interactions are weakened by electrostatic repulsion of negatively charged phosphate moieties [51,52]. The relatively low transition temperatures of anionic PG bilayers are largely attributable to such electrostatic repulsions, which is partially mitigated by the H-bonding interactions among the exchangeable protons of the glycerol headgroup and van der Waals interactions in the interfacial regions of the lipid bilayer [78,79,80].

Additionally, in the case of 1-palmitoyl-2-oleoyl-sn-glycero-3-phospho-(1′-rac-glycerol) (POPG) membranes, the peptide maintains its organization in two helices approaching on the surface of the bilayer (Appendix A). The side chain of W2 can penetrate deeply (see Figure 3G) as observed in the case of POPE, but the event is rarer (see Appendix A). The network of polar interactions reproduces what is observed with POPE, but in this case, the headgroup does not contain an amine and salt bridges involving E9 and D17 are inevitably lost. As a compensation, the number of interactions does not change because of the frequent involvement of R1 making many salt bridges and H-bonds with the membrane phosphate groups.

#### 2.5.2. The Effect of CXJ on PE/PG Mixtures Frequently Found in Bacterial Membranes

As stated above, PE and PG headgroups are frequently found in bacterial membranes. It should be stressed that PE and PG can form a strong electrostatic and H-bond networks favoring gel over fluid liquid-crystalline phases. PE amine group is the main H-bond donor that can interact with various acceptors in PE or PG. On the contrary, PG/PG H-bonds are rarely formed. Atom packing favors interactions between PE and PG over PE and PE. As a consequence, PE/PG bilayers are more difficult to disrupt than bilayers where inter-lipid H-bond cannot be formed [82,95]. Bacteria change PE/PG ratios of their membranes to control membrane permeability and stability [78]. The incorporation of peptides affects the interactions among lipids in two main ways. First, the presence of these peptides disrupts part of the hydrogen-bonding networks between PG and PE headgroups. Second, the polar/charged sidechains and both termini of the peptide compete with charged and H-bond forming groups on adjacent lipids. Due to the rich network linking PE and PG, any significant peptide-induced disruption has a considerably greater effect on its T_m_ than in PC and PG bilayers, where inter-lipid interactions are considerably weaker [78,79,80].

When analyzing the simulation with the mixture PE/PG (snapshot in Figure 3H), we observe the same type of polar contacts as observed for the pure components (Figure 5C) and an overall U-shaped structure (Figure 4D). However, a marked preference for PG headgroups (violet in Figure 5C) is apparent despite its lower amount (30%). This is likely due to the overall negative charge of POPG not present in POPE, which attracts the highly positively charged CXJ. Furthermore, the number of polar contacts occurrences raise significantly as compared to the pure components. In terms of van der Waals contacts (Figure 6C), more residues are allowed inside the bilayer compared with pure POPG (Appendix A). We believe that the smaller steric hindrance of PE with respect to PG facilitates the entrance of the peptide and consequently the establishments of polar contacts with POPG. At the same time, the reduction in the overall membrane negative charge (due to the presence of PE) limits the tendency of the peptide to remain at the level of phosphate moieties, allowing a deeper insertion.

#### 2.5.3. The Effect of CXJ on PE/PG/CL Mixtures and Pure CL Membranes

The interest in PE- and CL-containing membranes is not only related to bacterial membranes but also to mitochondria, whose inner membrane contains a high PE/PC ratio, PG and up to 25% of CL [96,97]. The similarity in composition is reminiscent of the bacterial origin of this organelle [98]. In the case of CXJ, the activity towards mitochondria is particularly important, as it could explain the induced apoptosis in cancer cells [5].

Figure 3I,J show that CXJ interacts strongly with the surface of both pure CL and PE/PG/CL membranes by means of helix I, maintaining its overall U-shaped fold for most of the trajectory (Figure 4E and Appendix A). In terms of polar contacts (Figure 5D and Appendix A), we observe the same network of interactions described for PE/PG membranes but a clear preference for CL in the mixed membrane PE/PG/CL (which constitutes only the 6% of the total lipid composition). Once again, this could be a purely electrostatic effect due to the doubly negative charge of CL with respect to PG (charge −1) and PE (charge 0). However, an important aspect is that CL may be seen as two PG phospholipids without headgroup, meaning that the access to phosphate groups is sterically facilitated [98,99]. This would also explain why CXJ penetrates quite deeply in such bilayers (Figure 6D and Appendix A) by means of multiple side chains in the pure CL membrane and that of W2 in the more realistic mixed model containing PE/PG and CL.

### 2.6. CXJ Interacts with Components of the Fungal Membrane

The antifungal activity of CXJ [1,2] and of about 39% of the members of the SR family (Figure 1) prompted us to investigate the behavior of CXJ in the presence of lipids typically found in fungal membranes. These are rich in PC and PE phospholipids but also ergosterol (ERGO). Phosphatidylinositol (PI) is often present, followed by PS, and it is generally found in the outer leaflet of fungal membranes, whereas in mammalian cells, it is mostly located in the inner leaflet [100,101,102,103,104,105,106,107]. With the exception of PC, CXJ interacts efficiently with bilayer composed of the most common fungal phospholipids: PE (Figure 3F, Figure 5B and Figure 6B) and PI (Figure 3K and Appendix A), partially conserving the U-shaped structure (Figure 4F). Due to the importance of sterols in membrane fluidity, we also decided to investigate the interaction with ergosterol and PE, one of the most common fungal lipids (Figure 3L and Appendix A). Appendix A shows that both polar and apolar (van der Waals) contacts are established. In the case of PI, CXJ interacts by means of lysine and arginine residues making salt bridges with the oxygen atoms of phosphate groups, while in the case of POPE, the amine of the headgroup is also involved, pointing to a more specific recognition.

### 2.7. The Effect of Concentration in the Activity of CXJ

AMPs often exert their antimicrobial activity by cooperativity. Multiple models have been proposed to explain how AMPs can destabilize biological membranes, some of them are carpet model, barrel-stave or toroidal (also called “worm-hole”) pore formation, detergent-type micellization, induction of non-lamellar phases, domain formation, non-lytic depolarization and localized thinning [108,109,110,111]. Another important feature of the membranes is that they can adjust their conformation to the environment modifying their shape and thickness accordingly [111,112], as described by the SMART model [113]. According to this model, designed antimicrobial compounds accumulate at the surface of the negatively charged membranes of bacteria or cancer cells. With increasing peptide concentration, transient micron-sized openings [108] can form in the membrane due to fluctuations in the local peptide-to-lipid ratio [111], allowing peptide translocation or the passage of other species [110,114,115,116].

In order to simulate a high concentration of peptides, we performed simulations in the presence of eight peptides. In particular, we monitored the effect on the fluidity of the membrane by calculating the order parameter of the palmitoyl chain (Figure 6) and the area per lipid (Figure 7). When analyzing the order parameter, the most apparent effect is the rigidification observed for membranes containing POPG (pure POPG, POPE/POPG and POPE/POPG/CL), which mimic bacterial membranes. Rigidification has been observed in cases where a strong electrostatic interaction is established [117], causing acyl chain packing [117,118].

On the contrary, an increased fluidification is observed in POPS and POPE membranes, caused by stochastic insertion of one of the eight peptides. This might indicate that the energetic barrier for the internalization might be lower in these cases. The phenomenon is rarely observed, and we believe that its characterization would require a much larger timescale, as discussed further in the text. Interestingly, PE and PS headgroups are both found in the external leaflet of cancerous cells, and PE is universally present in bacterial membranes.

Finally, we have analyzed the effect of the peptides on the area per lipid. This parameter allows us to monitor the effect on the curvature of the membrane [119,120]. In the absence of interacting species and in “planar” membranes the average area per lipid in both the upper and lower leaflets is virtually identical. On the contrary, this parameter changes in the two leaflets when a negative or positive curvature is produced. Appendix A shows how CXJ can create a significant negative curvature in its target membranes but not in membranes mimicking mammalian cells (POPC and POPC/CHO, Appendix A). The effect is particularly evident in POPS and POPG membranes (Appendix A) but also for all phospholipid components of bacterial membranes (POPE and CL, Appendix A). 

### 2.8. The Effect of C-Terminal Amidation in the Activity of CXJ

It has been shown that amidation of the C-terminus results in increased antimicrobial activity and even better performance in terms of cytotoxicity [3]. In the attempt to understand the molecular basis of such an effect, we repeated all the measurements for the amidated form (CXJN). Results are supplied as Appendix A.

First of all, it should be noted that the C-terminus is not essential for the antibacterial activity, as suggested by the SR family (Figure 1), in which truncated peptides 16 and 17 retain antibacterial properties [121]. Secondly, it should be noted that the absence of the terminal carboxylate should affect the interaction with bacterial and cancerous membranes in two ways: (i) the overall charge becomes more positive, thus increasing the Coulomb attraction to negatively charged membranes such as those of bacteria and cancer cells; (ii) we have shown that in the case of CXJ the terminal carboxylate moiety forms salt bridges with the amine of both PE and PS headgroup. Such an interaction may compete with the establishment of polar contacts between helix I and the membrane, reducing the overall affinity for target bilayers. These two factors alone might explain why amidation of the C-terminus has the effect of increasing the antibacterial activity of CXJ. However, a more detailed analysis of simulation might reveal more specific features.

In terms of structure, the contact maps (Appendix A) indicated a very similar behavior for CXJN compared to CXJ. In most cases, the U-shaped conformation approaching helix I at the N terminus and helix II at the C terminus is conserved.

When analyzing polar contacts (Appendix A), clear differences are apparent in the case of PS and PE membranes, for which the headgroups contain a protonated amine group. In the case of non-amidated CXJ, these amines are involved in occasional salt bridges with the terminal carboxylate (Figure 5A–D) that are obviously lost in CXJN, because the amidation prevents their formation. We did not observe other striking effects, probably because the helix II at the C-terminus did not show to interact stably with the membrane neither in the case of CXJ. In CXJN, the only effect is an increase in the involvement of K33 and K37 side chains (belonging to helix II) in the formation of salt bridges with oxygen atoms of the membrane.

When analyzing van der Waals contacts (Appendix A, we surprisingly observed a deeper insertion of CXJN in the membrane core, which also extends to the C-terminus in the case of pure POPS (Appendix A) and POPE (Appendix A). We hypothesized that the salt bridge formed between the terminal carboxylate and the amine of POPE (or POPS) attracts helix II to the membrane but also impedes its deeper internalization because of its strength; the repulsion to the negatively charged phosphate groups of the phospholipids may also contribute to preventing the internalization of the C-terminus. Both constraints are absent for CXJN, in which helix II can form salt bridges with oxygen atoms of the membrane by means of K33 and K37 side chains and descend deeper in the absence of electrostatic repulsions.

The insertion of CXJN in POPE and POPS membranes can also be monitored by the reduction in the order parameter of the lipid acyl chains (Appendix A). As discussed earlier, AMPs often increase this order parameter in a first phase (when polar contacts are established) and subsequently lower it when the peptide penetrates more deeply in the bilayers.

### 2.9. Final Remarks on the Internalization of CPPs

Many AMPs can cause a transient permeabilization of the membrane. In these cases, leakage starts shortly after peptides are added and subsequently slows down or stops. The leading hypothesis to explain this phenomenon is that the accumulation of the peptides in the outer leaflet of the membrane creates an imbalance of mass, charge, surface tension and lateral pressure that eventually leads to a stochastic local dissipation, causing the membrane to become transiently permeable [122,123]. Stochastic permeabilization allows CPPs to enter the cell without forming channel-like pores, a process typically requiring seconds to tens of seconds [122,123,124]. If binding and structural rearrangement occur quickly, a lag phase is caused by a higher energy barrier opposing translocation, probably originated by the hydrocarbon core. Some AMPs are able to lower this barrier and perturb the hydrocarbon core, a process depending on factors such as peptide concentration and temperature.

As opposed to common AMPs, many CPPs are able to enter the cell without damaging the membrane [7,8,32] and exert their killing action inducing apoptosis or targeting intracellular macromolecules such as DNA, RNA, ribosomes or organelles such as mitochondria [11,31,109,125,126,127,128], as it is the case for MPPs [11]. This mechanism is used by CPPs such as coprisin, some magainins or cecropins and postulated for CXJ [4,5].

MD simulations and modeling of membrane permeabilization rely on the assumption that permeabilization is an equilibrium process, a condition that is not always fulfilled, especially in the case of stochastic permeabilization [122,123,124]. The detection of such long processes [129,130] would require more advanced sampling algorithms including dual-resolution MD [131], coarse-grain simulations, steered MD [132], umbrella-sampling [133,134], metadynamics [135,136] or replica exchange, among others [130,137,138,139]. Our aim is to characterize the first steps of the interaction that are stationary on shorter timescales.

## 3. Materials and Methods

### 3.1. Sequence Alignment by ADAPTABLE Web Server

The family of peptides sequence-related to CXJ (RWKIFKKIEKMGRNIRDGIVKAGPAIEVLGSAKAIGK) was created by the family generator page of ADAPTABLE webserver (http://gec.u-picardie.fr/adaptable/) using “Create the family of a specific peptide” option with the following parameters: “anticancer = y”; “antibacterial = y”; “activity (µM) = 10”; “anticancer activity (µM) = 10”; “Substitution matrix = Blosum45”; “Minimum % of similarity = 51”. As ADAPTABLE continuously updates with new entries, sequence-related families might change slightly with the time [45].

### 3.2. Molecular Dynamics Simulations

Systems for simulations were prepared using CHARMM-GUI [140,141,142]. A total of 128 lipid molecules were placed in each lipid bilayer (i.e., 64 lipids in each leaflet) and peptide molecules were placed over the upper leaflet at non-interacting distance (>10 Å). Lysine and arginine residues were protonated. Initial peptide structure was obtained from Satpdb database (entry 12223 [49]). Amidation of the C-terminus was achieved, when desired, via the CHARMM terminal group patching a functionality fully integrated in the CHARMM-GUI workflow. In case of calculations with 8 peptides, they were placed next to each other but not in contact. A water layer of 50 Å thickness was added above and below the lipid bilayer, which resulted in about 15,000 water molecules (30,000 in the case of CL) with small variations depending on the nature of the membrane. Systems were neutralized with Na^+^ or Cl^−^ counterions.

MD simulations were performed using GROMACS software [143] and CHARMM36 force field [144] under semi-isotropic NPT conditions for bilayers [145,146]. The TIP3P model [147] was used to describe water molecules. Each system was energy-minimized with a steepest-descent algorithm for 5000 steps. Systems were equilibrated with the Berendsen barostat [148] and Parrinello–Rahman barostat [149,150] was used to maintain pressure (1 bar) semi-isotropically with a time constant of 5 ps and a compressibility of 4.5 × 10^−5^ bar^−1^. Nose–Hoover thermostat [151,152] was chosen to maintain the systems at 310 K with a time constant of 1 ps. All bonds were constrained using the LINear Constraint Solver (LINCS) algorithm, which allowed an integration step of 2 fs. Periodic boundary conditions (PBC) were employed for all simulations, and the particle mesh Ewald (PME) method [153] was used for long-range electrostatic interactions. After the standard CHARMM-GUI minimization and equilibration steps [145], the production run was performed for 1000 ns for CXJ in solution and in the presence of membranes, except for CXJN and simulations with 8 peptides (500 ns). The whole process (minimization, equilibration and production run) was repeated once in the absence of peptide and twice in its presence. Convergence was assessed using RMSD and polar contacts analysis.

All MD trajectories were analyzed using GROMACS tools [154,155] and Fatslim [156]. MOLMOL [157] and VMD [158] were used for visualization. Graphs and images were produced with GNUplot [159] and PyMol [160].

## 4. Conclusions

Current treatments for EC are based on surgical approaches combined with chemotherapy, but these strategies have poor outcomes. ACPs are good candidates in the search for new active compounds, as they can specifically target cancer cells, thus providing new ways to overcome the toxicity of chemotherapy. As compared to currently used drugs, they are intrinsically less prone to develop distance because of their ability to destabilize or bypass biological membranes, whose composition cannot be changed by a single point mutation [31]. An important feature of CXJ is its ability to penetrate the cell and target mitochondria, opening a wide range of possibilities to develop targeted therapies inducing apoptosis in cancer cells or act as antimicrobials in intracellular infections, including viruses. Deciphering the mechanism of action of CXJ is, therefore, essential to engineer its action for targeted therapies.

In this work, we have shown how the CXJ peptide, largely unstructured in solution, assumes alpha helical conformation in the presence of biomimetic membranes. Two helices (helix I, from residues 1 to 22, and helix II, from 25 to 37) can be distinguished, separated by a short loop containing P24. While sequence alignment and MD suggests that helix II might not be essential for the activity, amidation of the C-terminus seems to increase the antibacterial properties of CXJ, probably due to the increase in the net positive charge of the peptide.

While not interacting with models of mammalian bilayers composed of PC and cholesterol, CXJ is able to specifically recognize PS and PE headgroups, both characterizing the outer leaflet of cancer membranes. These phospholipids play an important role in resistant mechanisms of fungi such as *Cryptococcus neoformans* or *Candida albicans* [100,101,161,162] and in the virulence of parasites such as *Plasmodium* [161] or other intracellular pathogens such as *Brucella* [163]. CXJ specifically interacts with the carboxylate atoms of PS by means of the side chain NH_2_ groups of arginine 1, 13 and 16. Coupling of these interactions with lipid flip-flops might explain how CXJ could act as CPP and penetrate the membrane of cancer cells without affecting its integrity [8,66,67,68]. Alternatively, CXJ could exploit transient permeabilization induced by high peptide accumulation on the membrane surface. With the PE headgroup, also present in bacterial and fungal bilayers, CXJ can interact via E9, D17 and E27, forming salt bridges with its protonated amine. At the same time, the side chains of R1,13 and K3,6,10 interact with phosphate moieties. In both cases, W2 deeply inserts its aromatic ring into the membrane core.

CXJ displays an important activity against many bacteria in the WHO priority list [164], such as *Staphylococcus aureus*, *Klebsiella pneumoniae* and *Pseudomonas aeruginosa*. In the case of bacterial-like membranes, the main interactions involve the formation of salt bridges with phosphate moieties of phospholipids. However, CXJ shows a marked preference for CL, followed by PG and PE. Since CL is abundant in mitochondria, this finding might explain its MPP properties and its apoptotic effect.

This analysis confirms the importance of the three conserved motifs (RWK, KKIEK and GIVKAGPA) that were highlighted by sequence-property alignment. The first contains the important tryptophan residue, which was found to deeply insert in target bilayers; the second resides in helix I and establishes frequent salt bridges with phosphate oxygen atoms by means of lysine residues and with the amine of PE or PS headgroups by means of the glutamate; the third breaks the helical structure separating helix I from helix II and provides interhelical mobility. Arginine residues in position 1, 13 and 16 are also conserved and may act as anchoring points specifically in PS containing membranes. The integration of biological activity with the analysis of contact map seems to suggest that arginine residues could be responsible for the CPP character of CXJ, while the lysine residues could account for its activity towards mitochondria. CXJ can in fact penetrate cancer cells exposing PS in their outer leaflet and destabilize mitochondrial membranes rich in PG and CL

## Figures and Tables

**Figure 1 ijms-22-00691-f001:**
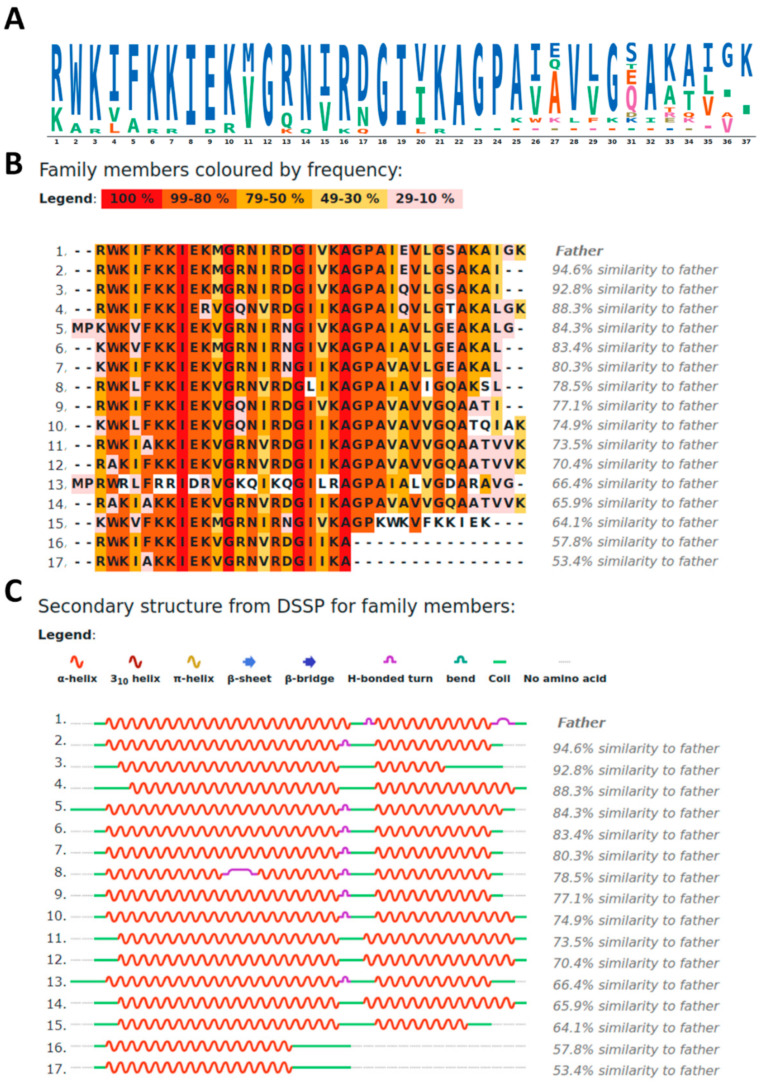
(**A**) Sequence logo calculated from cecropinXJ (CXJ) sequence-related (SR) family; (**B**) members of CXJ SR family; (**C**) Define secondary structure of proteins (DSSP)-based secondary structure for each member of CXJ SR family.

**Figure 2 ijms-22-00691-f002:**
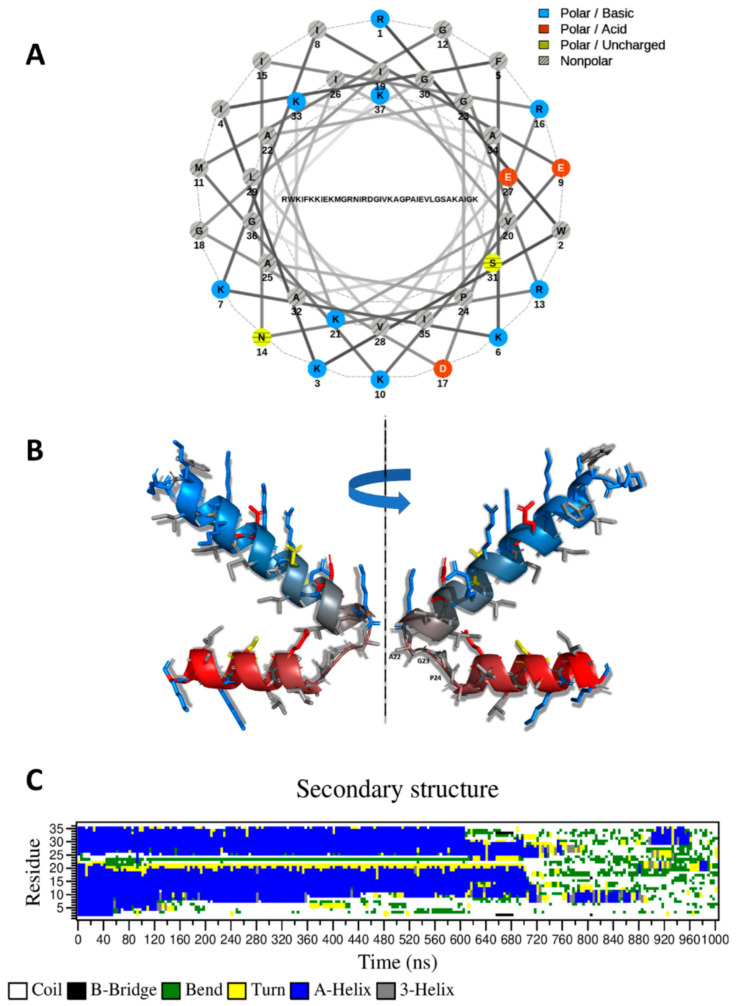
(**A**) Helical wheel plot (generated with NetWheels [50]); (**B**) I-TASSER structural prediction of CXJ. The peptide backbone is colored from blue (N-terminus) to red (C-terminus). Side chains are shown as sticks with the following color code: positively charged (blue), negatively charged (red), non-polar (light gray), polar (yellow). Residues separating the two helices are labeled; (**C**) DSSP secondary structures calculated along molecular dynamics (MD) simulation of CXJ in solution.

**Figure 3 ijms-22-00691-f003:**
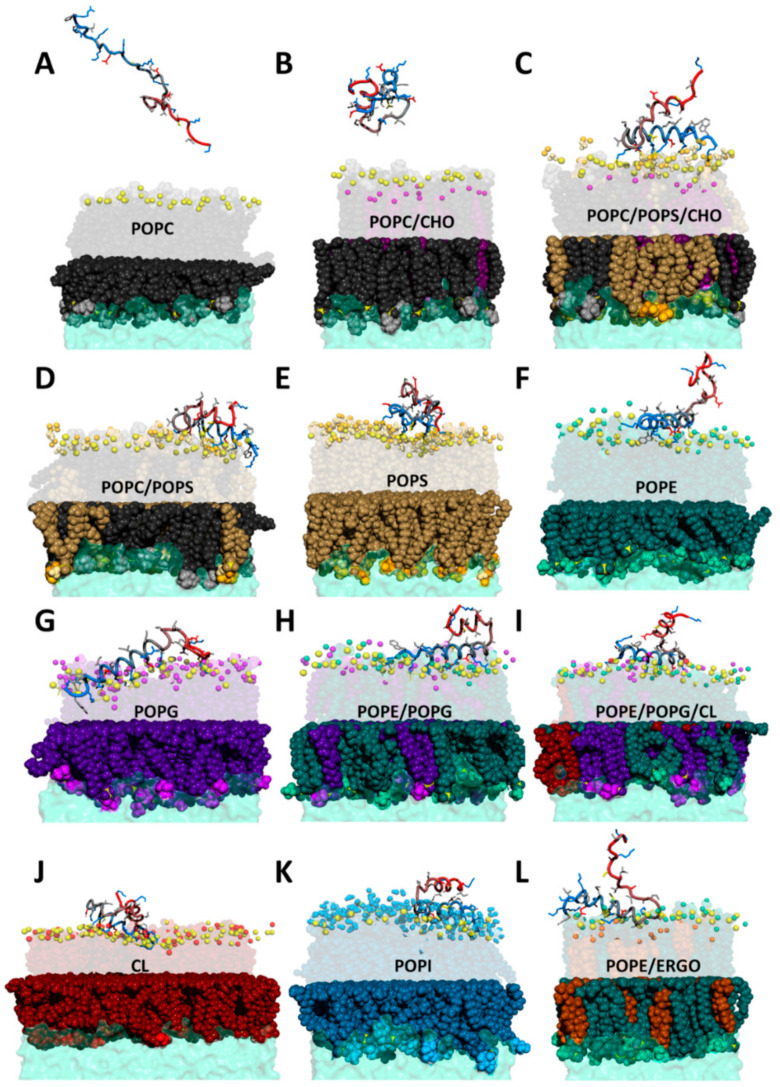
MD snapshots representative of CXJ peptide interacting with several membranes of variable phospholipid compositions. (**A**) 1-palmitoyl-2-oleoyl-glycero-3-phosphocholine (POPC); (**B**) POPC/cholesterol (CHO); (**C**) POPC/1-palmitoyl-2-oleoyl-sn-glycero-3-phospho-L-serine (POPS)/CHO; (**D**) POPC/POPS; (**E**) POPS; (**F**) 1-palmitoyl-2-oleoyl-sn-glycero-3-phosphoethanolamine (POPE); (**G**) 1-palmitoyl-2-oleoyl-sn-glycero-3-phospho-(1′-rac-glycerol) (POPG); (**H**) PE/PG, (**I**) PE/PG/cardiolipin (CL); (**J**) CL; (**K**) 1-palmitoyl-2-oleoyl-sn-glycero-3-phosphoinositol (POPI); (**L**) PE/ergosterol (ERGO). Color code: phosphorus atom: yellow, POPC black (body) and light gray (choline group), POPS brown (body), gold (headgroup), light yellow (amine of the headgroup) and orange (carboxyl of the headgroup), POPE dark green (body), turquoise (headgroup), light green (amine of the headgroup), POPG dark violet (body), violet (headgroup), light violet (hydroxyls of the headgroup), POPI blue (body), light blue (headgroup), cyan (hydroxyls of the headgroup); CL dark red (body) and light red (headgroup), ERGO dark orange (body) and light orange (hydroxyl); CHO purple (body) and light purple (hydroxyl). Panel (**B**–**D**,**H**,**I**,**L**) show lipid composition modeling mammal cells (**B**), cancer cells (**C**,**D**), bacteria (**H**,**I**) and fungal cells (**L**). For clarity, only functional groups of headgroups are shown (spheres) in the upper leaflet. CXJ peptide is shown as a “tube” colored from blue (N-terminus) to red (C-terminus). Side chains are shown as sticks with the following color code: positively charged (blue), negatively charged (red), non-polar (light gray), polar (yellow).

**Figure 4 ijms-22-00691-f004:**
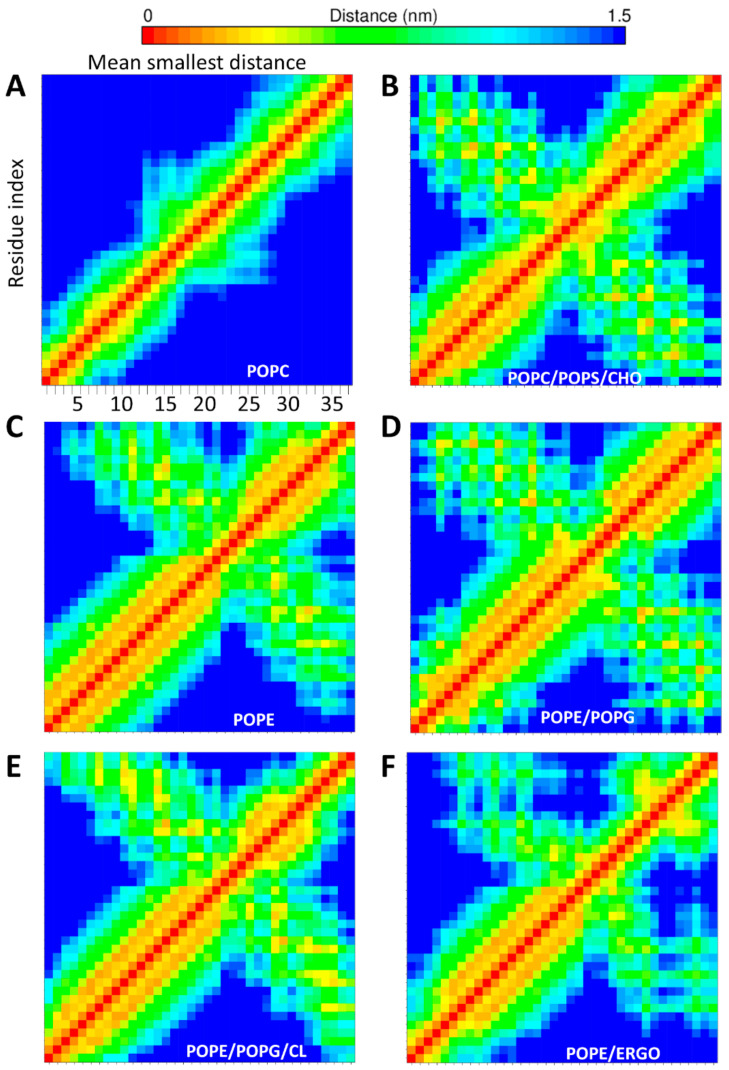
Contact maps showing how each residue of CXJ interacts with others in presence of different membranes: (**A**) POPC; (**B**) POPC/POPS/CHO; (**C**) POPE; (**D**) POPE/POPG; (**E**) POPE/POPG/CL; (**F**) POPE/ERGO.

**Figure 5 ijms-22-00691-f005:**
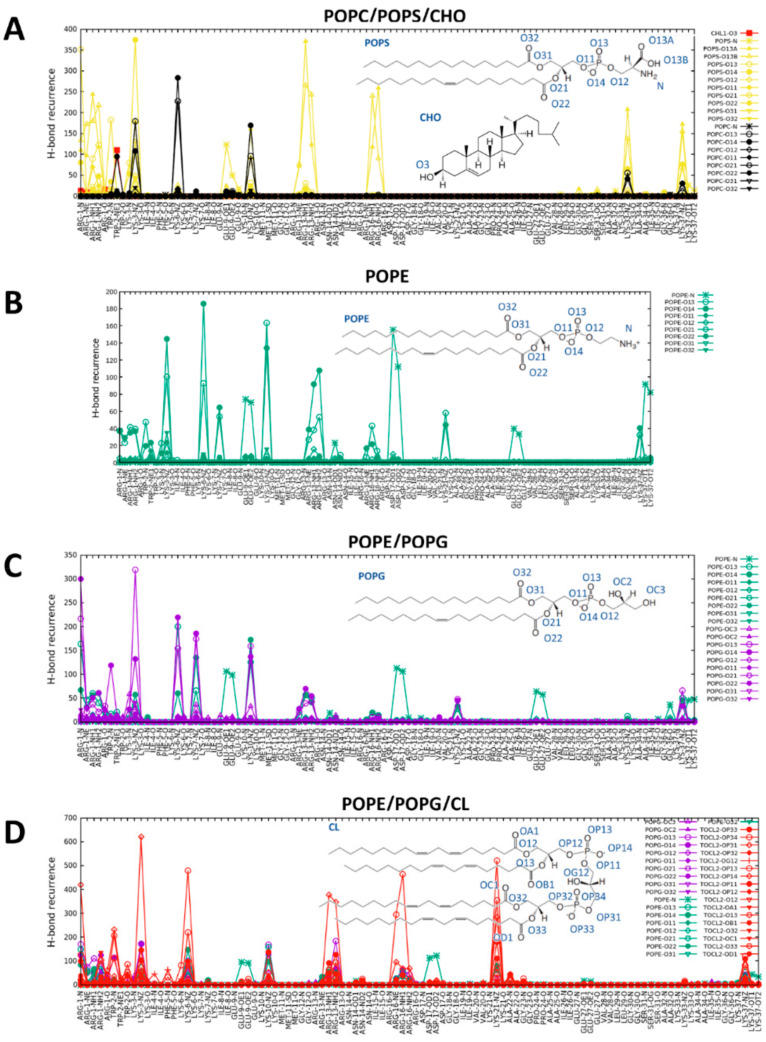
Occurrence of polar atom contacts (H-bonds and salt bridges) between CXJ peptide and various membrane bilayers calculated along MD simulation trajectories: (**A**) POPC/POPS/CHO; (**B**) POPE; (**C**) POPE/POPG; (**D**) POPE/POPG/CL. TOCL2 refers to CL.

**Figure 6 ijms-22-00691-f006:**
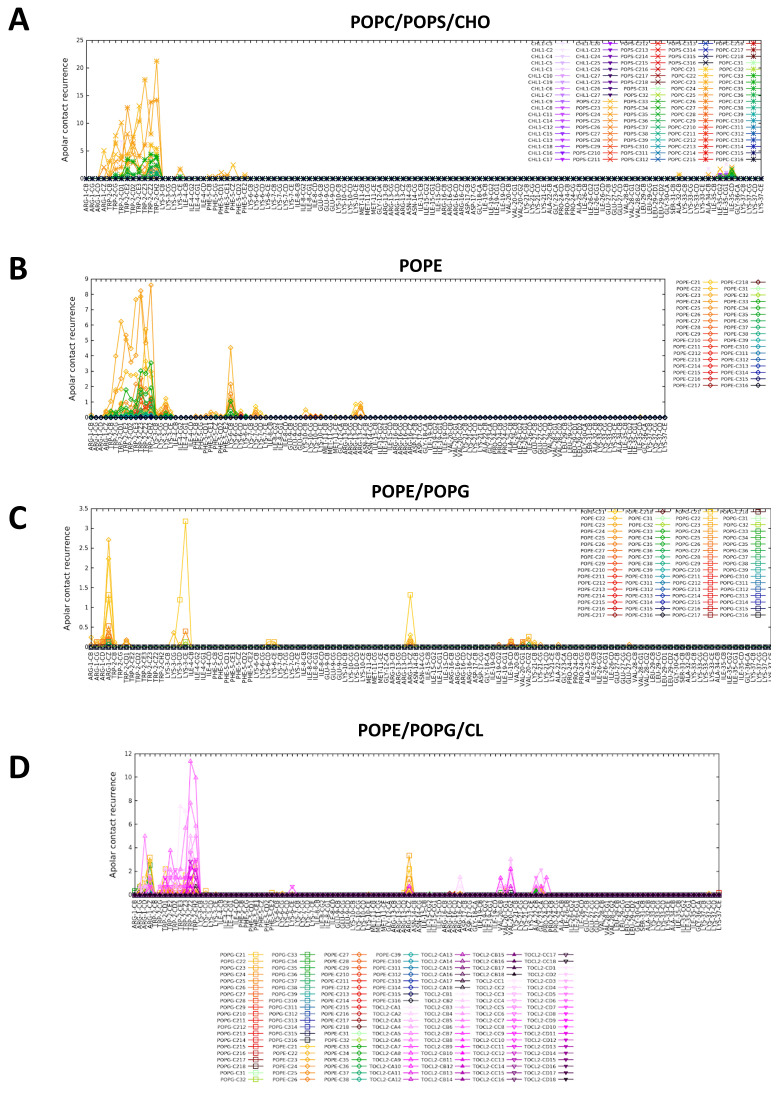
Occurrence of van der Waals contacts between CXJ peptide and various membrane bilayers calculated along MD simulation trajectories: (**A**) POPC/POPS/CHO; (**B**) POPE; (**C**) POPE/POPG; (**D**) POPE/POPG/CL. TOCL2 refers to CL.

**Figure 7 ijms-22-00691-f007:**
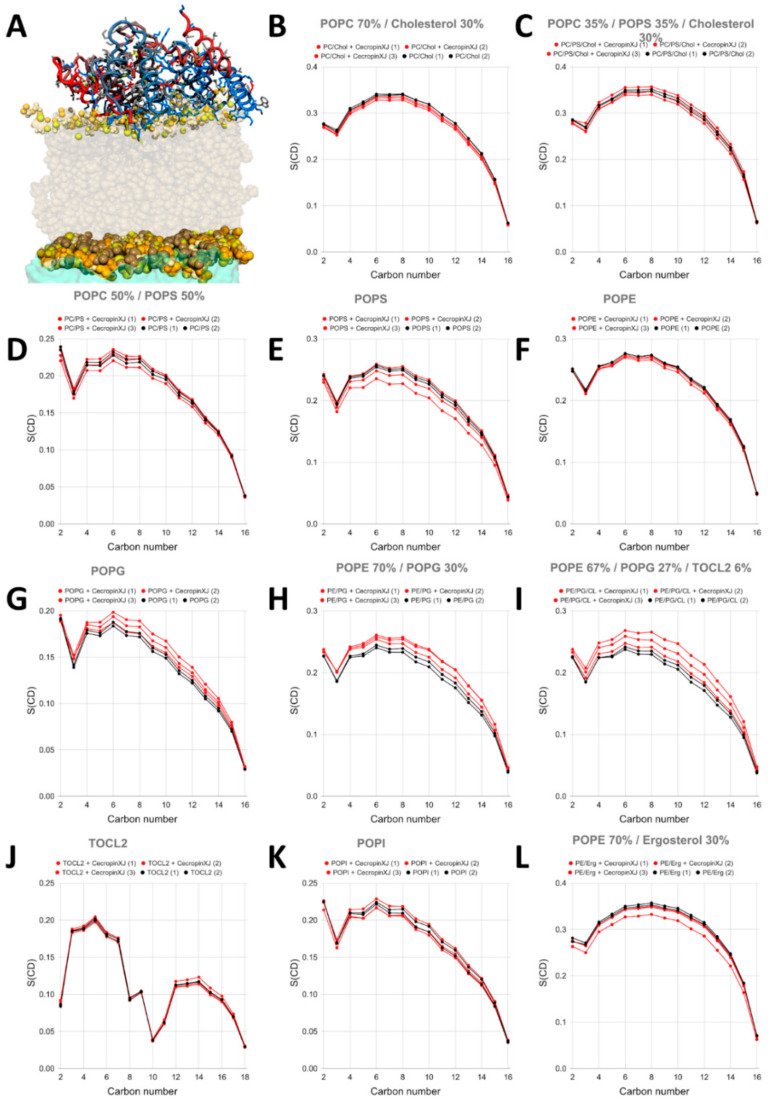
Order parameter of C-H moieties of palmitoyl side chains in membranes containing various phospholipids compositions as calculated from multiple repetitions of MD simulations in the absence (2 repetitions in black labeled as 1 and 2) and in the presence (3 repetitions in red labeled from 1 to 3) of eight CXJ peptides. (**A**) example snapshot of one simulation with eight peptides (color code in the caption of Figure 3); (**B**) POPC/CHO, (**C**), POPC/POPS/CHO, (**D**) POPC/POPS, (**E**) POPS, (**F**) POPE; (**G**) POPG; (**H**) POPE/POPG; (**I**) POPE/POPG/TOCL2; (**J**) TOCL2; (**K**) POPI; (**L**) POPE/ERGO. TOCL2 refers to CL.

## Data Availability

Data is contained within the article or Appendix A.

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
