# Peer review of "Molecular Basis of the Anticancer and Antibacterial Properties of CecropinXJ Peptide: An In Silico Study"

_ijms, 2021, doi:10.3390/ijms22020691_

Round 1

Reviewer 1 Report

This paper describes the action mechanism studies of antimicrobial and anticancer peptide Cecropin XJ by MD simulation and sequence-property alignment. The methods used and the results are solid and interesting, in particular, analysis of the peptide interaction with various types of biomembranes well supported the biological character of this peptide. The results reported in this paper would pave the way to more development and future progress of this therapeutically important and useful peptide. Accordingly, this paper deserves to be published in this journal.

Reviewer 2 Report

The Authors present the manuscript entitled "Molecular basis of the anticancer and antibacterial properties of CecropinXJ peptide. This paper if very interesting, also much work was done to provide such information. Nevertheless, some minor corrections need to be provided before this paper is published.

  1. In my opinion, the title of the manuscript is somehow confusing and neet to be changed a bit. The entire work presented in the manuscript is based on in-silico studies, and is an assumption rather than evidence. Therefore such information should be provided. Otherwise, the reader will search in vain for the results of in vitro tests that support the title indicated.
  2. Please provide the definition for abbreviations used in the subtitle 2.3
  3. In the subtitle 2.1 the Authors written that "two helices can be distinguished ...by the AGPA..." Then the Authors divide the sequence into two helices- Helix I (1-21), and Helix II (25-37). However, 25 residue is for Alanine which is situated in the AGPA motif. Thus, Helix II should be 26-37, no?
  4. On what basis did the Authors choose the ratios of bacteria phospholipids? Was the selection made according to any specific algorithm (pH, water content, etc?) or maybe on the basis of the literature data?
  5. In line to this, is it possible that the environmental factors due to which the bacteria changes its phospholipid ratio, may directly affect the interactions between the drug and the bacteria, and thus the mechanism of drug activity?
